# Extended Continuous Cooling Transformation (CCT) Diagrams Determination for Additive Manufacturing Deposited Steels

**DOI:** 10.3390/ma15093076

**Published:** 2022-04-23

**Authors:** Kristýna Halmešová, Radek Procházka, Martina Koukolíková, Jan Džugan, Pavel Konopík, Tomasz Bucki

**Affiliations:** 1COMTES FHT a.s., Průmyslová 995, 33441 Dobřany, Czech Republic; rprochazka@comtesfht.cz (R.P.); mkoukolikova@comtesfht.cz (M.K.); jdzugan@comtesfht.cz (J.D.); pkonopik@comtesfht.cz (P.K.); 2Department of Metallurgy and Material Technologies, Kielce University of Technology, Al. Tysiąclecia Państwa Polskiego 7, 25314 Kielce, Poland; tbucki@tu.kielce.pl

**Keywords:** mini-tensile test, continuous cooling transformation diagram, laser directed energy deposition, selective laser melting

## Abstract

Continuous cooling transformation (CCT) diagrams are widely used when heat treating steels and represent which type of phase will occur in a material as it is cooled at different cooling rates. CCT diagrams are constructed on the basis of dilatometry measurements on relatively small testing samples (cylindrical shape with diameter of 4mm and length of 11 mm in this study). The main aim of this work was to demonstrate the possibility of evaluating the tensile test properties using mini-tensile tests from miniature volumes (1.4 × 10^−7^ m^3^ for one sample) subsequent to determination of the CCT diagram and to extend a standard CCT diagram with information about strength, ductility and the estimated value of the work-hardening coefficient. Mini-tensile tests (MTT) were recently developed due to the low availability of experimental material and have already been successfully used for local mechanical property characterization of metals. CCT diagrams were constructed for 42CrMo4 steel prepared by the laser-directed energy deposition (L-DED) process, for commercially available 42CrMo4 steel conventionally manufactured (for comparison of traditional processing and AM preparation) and for H13 tool steel deposited by the selective laser melting (SLM) process.

## 1. Introduction

Continuous cooling transformation (CCT) diagrams are broadly used to design specific heat treatments and to predict the microstructure and mechanical properties after thermal treatments [1]. Most of these curves are determined by dilatometry with or without deformation prior to cooling [2]. CTT diagrams are generally more appropriate for engineering applications as components are cooled (air cooled, furnace cooled, quenched etc.) from a processing temperature as this is more economic than transferring to a separate furnace for an isothermal treatment. During the designing of CCT diagrams for steels, the samples are austenitized and then cooled at predetermined cooling rates and the degree of transformation is measured. The locations and shapes of the supercooled austenite transformation curves, plotted on the CCT diagrams, depend mostly on the chemical composition of the steel, extent of austenite homogenizing, austenite grain size, as well as on austenitizing temperature and time [3]. Experimental elaboration of the CCT diagram is time consuming and requires expensive testing equipment. Therefore, many attempts of modelling austenite transformations in the steel during cooling are being undertaken to be able to calculate the CCT diagrams from the chemical composition of the steel and its austenitizing temperature [4]. Calculation methods give an alternative to experimental measurement in providing the material data required for heat treatment process simulation but in many cases significantly differ from experimental results.

Except for the developed microstructures after cooling, the CCT diagrams also provide information about the hardness of the quenched material. The mechanical properties are usually measured separately from the CCT samples that need an additional experimental material. 

In order to avoid the need for additional material processing and to ensure the same and well-defined heat treatment conditions, a previously developed mini-tensile test (MTT) method is applied here for tensile properties’ determination from dilatometry samples. MTTs [5,6,7,8,9,10,11] can be beneficial for the quality and reliability assessment of the final manufactured part, for the determination of local properties of critical parts, for example, sharp bending locations with small radiuses where residual formability and elongation can be near to the material limit [5]. The potential advantages and limitations of using MTT for local characterization of additively manufactured samples and components are discussed in more detail for additively manufactured Ti6Al4V in [10]. Very good agreement was found for cold rolled steel for drawing and cold forming (DC01-DC06) with the use of standard-sized and MTT specimens [11]. MTTs can play a very important role in material research where there is a limited amount of experimental material (additive manufacturing, development of new material), for material parameter determination for various material states after production or postprocessing such as: SPD processes (HPT, ECAP), miniature products (dental implants, mini-grooves in electronics), welds, etc.

Until recently, the hardness measurement was the only testing method with an acute lack of experimental material, for example, in the case of a rotating target neutron source [12]. There is an accepted theory for the correlation of the indentation hardness and strength proposed by Cahoon [13]. The relation between the hardness *H* and the ultimate tensile strength *σ_u_* based on the experimental results and the strain hardening behavior of the material may be given as:(1)$$\sigma_{u}=\frac{H}{2.9}{\left(\frac{n}{0.217}\right)}^{n}\;,$$
where *n* is the strain hardening exponent of the given material. The possible values of *n* range from *n* < 0.1 (work-hardened material) to *n* ~ 0.5 (fully annealed material) or typical values of *n* tend to be in the range 0.15–0.18 for high-strength alloy steels and in the range of 0.20–0.23 for low carbon steels [14].

Since then, many authors have attempted to relate mechanical properties and hardness or microhardness to all kinds of materials and it has been demonstrated that in the case of metals and alloys, there is a quite precise correlation between hardness and flow stress [15]. Abson et al. [16] studied the hardness–strength correlation for some titanium-based alloys and found that a unique correlation exists but it is not the one predicted by Cahoon in Equation (1). So it is generally agreed that hardness correlates well to ultimate tensile strength and only loosely to yield strength and ductility with a lot of scatter. 

Where stress–strain data are not available, the work-hardening coefficient *n* may usually be estimated from the equation
(2)n=(m−2)
where *m* is the Meyer hardness coefficient. Equation (2) has been shown to be true theoretically, and its validity has been demonstrated for a variety of materials. The quantity *m* is obtained as the slope of a log–log graph of load *W* and indentation diameter for a spherical indenter [16].

It was seen that without having stress–strain data it is not clear which work-hardening coefficient *n* to use in Equation (1). It was also shown that the strain hardening exponent varies from alloy to alloy and also depends on the condition of the material (whether it has been plastically deformed, heat treated, etc.) [14], on grain size and can be influenced by processing, test temperatures as well as the strain rate that was used during the tensile tests [17]. This feature gives significant inaccuracies in the estimation of mechanical properties and may be applied where tensile data are required but where tensile testing is impossible, impractical and/or desirable. Nevertheless, in a tensile test, not only information about strength is obtained but also about elongation and it also appears that hardness testing cannot differentiate between brittle and ductile behavior of the material. There are no guarantees that if the hardness is within the specification, then the other mechanical properties of concern are also likely to be within the specification. For these reasons, it is desired and more advantageous to use MTTs in cases where it is possible. With the help of MTT samples, it is possible to obtain not only tabulated tensile strength characteristics, but also stress–strain curves for other processing purposes. These curves are important in describing behavior during plastic deformation in terms of deformation, strain rate and temperature, for example, setting nonlinear constitutional equations of plasticity and to determine the feasibility and effectiveness of metal processing processes.

At present, there has not been published anywhere any similar approach where dilatometry cylinders have been used directly for the determination of tensile properties. There is very little awareness of this possibility. Perhaps, this is why dilatometry samples have been used only for metallographic analyses and mechanical tensile properties are estimated only on the basis of correlations with microhardness. A lot of work showing comparisons of tensile data for standard size and MTT specimens has been published [7,8,9,10]. So, in this work the main aim was to demonstrate the possibility of using MTT from the dilatometry cylinders with dimensions D4 × 11 mm to evaluate the tensile properties subsequent to determination of the CCT diagram. This gives an opportunity for the development of metal alloys on a microscale. This procedure, in comparison with the standard procedure containing only metallographic analyses and microhardness measurements, aims to obtain additional reliable information at the level of standard samples for tensile tests (STT) while reducing the demands on the volume of experimental material and instrumentation. This process results in an extended CCT diagram also providing the strength, ductility characteristics and estimation of the work hardening exponent *n*. 

In this study, a demonstration of this approach is presented for 42CrMo4 steel prepared by the laser-directed energy deposition (L-DED) process (designated as 42D), for commercially available 42CrMo4 steel conventionally manufactured (for comparison of traditional and AM preparation) (designated as 42C) and for H13 tool steel deposited by the selective laser melting (SLM) process (designated as H13SLM).

## 2. Materials and Methods

The chemical composition of commercially available 42C steel was determined using a Brucker Q4 Tasman optical spectrometer (BRUKER AXS Inc., Madison, WI, USA). The composition of both powder steels presented corresponds to the manufacturer’s data sheets. The results are summarized in Table 1. 

Steel for quenching and tempering 42CrMo4 is widely used in automotive and aircraft components which require high toughness. H13 tool steel is commonly employed in the injection molding industry, because it offers high strength at elevated temperatures and high hardness. It also has the advantages of high resistance to thermal shock and thermal fatigue, high abrasion resistance and heat resistance. In SLM applications, H13 can also achieve in situ hardening during the process, which can potentially reduce the need for postbuild hardening heat treatments [18]. Material here signed as 42D was produced by commercially available L-DED system (MX-600, Insstek, Daejeon, Korea) with 2 kW Yttrium fiber laser and powder provided by Sandvik was used. Argon gas was used as a protective atmosphere during the deposition process. After L-DED deposition no other thermal treatment was applied. Using these additive manufacturing techniques the cylinders with the diameter of 4 mm and length of 11 mm were produced and no additional mechanical machining except the removal of the building plate was needed. H13 SLM was processed using a SLM280HL machine (SLM Solution AG, Lübeck, Germany). The machine is equipped with an Yb:YAG laser (IPG Photonics) with 400 W nominal power. The laser power was set to 175 W and scanning speed was 610 mm/s. Inner part of the specimen was built with stripe hatch strategy. The laser tracks were rotated 90° for each layer in order to minimize the inner porosity. Nitrogen gas was used as a protective atmosphere. To minimalize the residual stresses in the samples, the substrate plate was preheated to 200 °C and building process was followed by annealing at 600 °C for 2 h.

A quenching dilatometer (RITA L78, Linseis, Selb, Germany) was used for dilatometry measurements with cooling rates ranging from 100 to 0.1 °C/s in helium atmosphere. Heating rate of 10 °C/s was applied for all samples. CCT diagrams were constructed on the bases of these measured cooling temperature dilatometry curves by so called “lever rule”, because the length change during the transformation reflects the ratio between transformed and untransformed phase. The transformation start temperature was determined as temperature at which 1% of austenite is transformed into a new phase and transformation end temperature at which 99% of transformed austenite was achieved.

The specimens for dilatometry measurements had diameter of 4 mm and length of 11 mm and for each cooling rate two samples were tested. The second one served for confirmation of the first one. From these specimens, mini-tensile samples for tensile testing were machined according to the geometry shown in Figure 1a. Due to the fact that there is a need to prepare as many samples as possible from a small amount of material (3 MTT specimens from each dilatometry cylinder) an unconventional machining process known as electric discharge machining (EDM) was used. This process results in a thin kerf cut and smooth cut surface covered by a thin oxide film. The manufacturing process consists of two steps. The first one includes EDM of the samples’ outer contour and then the slicing of first two samples. Samples finished by this procedure were further sanded with abrasive brushing with grit P600 to remove the oxide film that was formed due to the EDM process. The second step includes a wet sanding of the third sample in order to reduce its final thickness to 0.5 mm. In this case, both faces of the sample were finished by wet sanding which results in a high quality surface. The surface quality of such small samples is crucial for the measurement accuracy of tensile test results [19,20].

The mini-tensile test methodology was developed to reliably measure tensile material characteristics from a minimum amount of experimental material [5]. The principle of the test is based on the standards EN ISO 6892-1 [21]. However, as these standards do not consider the testing of miniature samples, an appropriate procedure was created. This methodology maintains the recommended test strain rate $\dot{\varepsilon }$ given by the equation:(3)$$\dot{\varepsilon }=\frac{v}{L}$$
where *v* is the actuator velocity and *L* is the length of sample’s parallel section. The relationship shows that the ratio of the parallel length section and the piston velocity of the test machine affects the strain rate $\dot{\varepsilon }$. To maintain the quasi-static test conditions, the piston velocity needs to be significantly reduced in comparison to a standard size sample. This brings some difficulties for machine control. Therefore, the mini-tensile samples were tested with the use of the specially designed LabControl tester with a capacity of 5 kN. In addition, it is necessary to ensure the sufficient sensitivity of the load cell, which is required for small specimen cross-sections (approx. 1 mm^2^). The deformation tracking while sample loading was provided by digital correlation system Mercury RT high-speed system (Sobriety, Kuřim, Czech Republic). The deformation was evaluated based on a digital image correlation (DIC) technique [22,23,24,25,26]. A necessary spot pattern on each specimen surface was made by the combination of black and white spray color applied by airbrush. A sufficiently fine pattern provides a very high accuracy of deformation measurement. It ranges in the order of micrometers. All tensile tests were performed at room temperature at strain rate of 10^−3^ s^−1^. In the final CCT diagram, there is one representative tensile curve displayed for clarity. For each testing regime, the average of three measurements and standard deviation were calculated.

One standard tensile test (STT) was also performed to demonstrate the suitability of microtensile tests (MTT). STT was carried out according to EN ISO 6892-1 with specimen geometry illustrated in Figure 1b. Both tests were carried out for 42CrMo4, for which the material was available for large test specimens. The material was subjected to heat treatment consisting of heating to 1000 °C with holding for 1 h followed by water quenching and tempering at 600 °C for 2 h followed by cooling in still air. This temperature regime resulted in the tempered martensite microstructure having the grain size 8.5. 

The Vickers microhardness HV1 was measured on a Struers DuraScan-50 (Struers A/S, Ballerup, Denmark) hardness tester according to ISO 6507—1: Vickers hardness measurement. For each sample, the microhardness was performed at three different positions, average value and standard deviation were calculated.

To validate each material phase and complete CCT diagrams, metallography of all samples was also studied. The microstructures were etched with Villela-Bain chemical agent and then examined using a light microscope NIKON ECLIPSE MA200 (Nikon, Tokyo, Japan). Fractography was performed on the scanning electron microscope JEOL 6380 (Jeol Ltd., Tokyo, Japan) under the secondary electron regime. The local chemical composition was measured by EDX analyzer INCAx-sight (Oxford Instruments, Abingdon, UK).

## 3. Results

The tensile tests results obtained for standard (STT) and mini geometries (MTT) from the same base material for 42CrMo4 (42C) are shown in Figure 2. The tensile curve obtained for the STT geometry (solid lines) was compared with MTT curves (discontinuous lines). 

Conventionally manufactured steel for quenching and tempering 42C and its CCT diagram together with the microhardness measurement, microstructure, tensile test records and also the mechanical properties determined from MTT directly from the dilatometry sample are displayed for five cooling rates (0.1, 1, 5, 10 and 50 °C/s) in Figure 3. The same work was done for the same steel 42CrMo4 prepared with DED process and the final CCT diagram is presented in Figure 4. All this information was achieved from five samples with a volume of only 1.4 × 10^−7^ m^3^ each. (Note that for one standard tensile sample more 1 × 10^−5^ m^3^ is required). Letters P/F, B, M indicate areas of austenite transformation to perlite with ferrite, bainite, martensite, respectively. Grain size stated in Figure 3 and Figure 4 refers to grain size of austenite that existed prior to its transformation to individual phases. 

The summary of the transformation temperatures for the individual cooling rates measured for conventionally prepared (42C) and DED prepared 42CrMo4 steel (42D) is expressed in Table 2.

Table 3 and Table 4 recapitulate the results of the mechanical properties determined from engineering stress–strain curves of MTTs for 42C and 42D steel. Along with this, the tables contain the results of the work-hardening exponent, *n*, which was calculated based on the true stress–true strain curve obtained for samples that reached at least a strain of 0.015 and the estimated values are also displayed in CCT diagrams in Figure 3 and Figure 4.

Additionally, Table 5 shows the results of microhardness measurement HV1 for 42C and 42D steels.

Figure 5 displays CCT diagram for H13 tool steel prepared with SLM extended with the mechanical properties determined from the tensile tests directly from the dilatometric samples. This CCT diagram consists of three cooling rates in the range of 1–100 °C/s.

## 4. Discussion

The results in Figure 2 proved that the tensile curves for MTT samples reveal almost identical curve shapes and tensile properties in comparison with STT sample geometry. Numerically, for 42CrMo4, the yield strength (Rp0.2) reached 942 MPa for the STT sample and 899 MPa ± 25.2 MPa for MTT samples, the ultimate tensile strength (Rm) was 1070 MPa and 1059 MPa ± 8.7 MPa for STT and MTT samples, respectively. A slightly lower value of Rp0.2 for MTT samples can be explained by a different scale of both samples which may be related to the local (sample orientation) and global strain (bulk material). Similar behavior was observed by Chen et al. where the effect of the volume ratio of surface grain to internal grain on the flow stress was investigated [27]. Furthermore, the elongation (A) of STT is in good agreement with MTT where the difference between STT and MTT is 6%. Based on these experiments, it can be concluded that the results obtained using MTT are credible and can be used for the determination of tensile properties.

From the CCT diagrams in Figure 3 and Figure 4, it is clear that the 42CrMo4 steel undergoes perlitic, bainite and martensite transformation according to the cooling rates (areas of transformation are symbolized with letters P, B and M). As the cooling rate increases from 0.1 to 50 °C/s, the perlitic microstructure changes via bainite to a martensite microstructure. When comparing the constructed phase transformation curves of the CCT diagrams in Figure 3 and Figure 4, it can be deduced that the CCT diagrams did not differ significantly from each other. More precisely, for individual cooling rates the phase transformation temperature (e.g., martensite start temperature, Ms, bainite start temperature, Bs, bainite finish temperature, Bf, etc.) do not differ from each other by more than 10%. This might be attributed to differences in chemical composition, because the alloying elements in steel generally affect the phase transformation kinetics and in the case of diffusional transformations, the transformation temperature also depends on the prior thermal history during manufacturing or on the grain size [28].

As expected, the perlite structures of 42C and 42D at a cooling rate of 0.1 °C/s exhibited the lowest ultimate tensile strength, approx. (816 ± 7) MPa ((686 ± 1) MPa for 42D) with average elongation reaching (22 ± 1)% ((18 ± 1) % for 42D). The cooling rate of 1 °C/s led to a bainite microstructure with ultimate tensile strength of (1365 ± 7) MPa ((1005 ± 6) MPa for 42D) and average elongation of (7 ± 1)% ((11 ± 0)% for 42D). Predominantly martensite microstructures at cooling rates of 5–50 °C/s demonstrate brittle behavior with a high value of ultimate tensile strength characterized by almost no observed elongation and the samples at these cooling rates did not reach a maximum load which could be identified as the ultimate tensile strength. The testing of very strong brittle materials using a tensile test method becomes more complicated. For this purpose, proper specimen geometry and a very accurate clamping device must be employed. Jirkova et al. successfully used a special subsize specimen geometry for tensile testing in the study of the mechanical properties of a tool steel with hardness of 800 HV [29]. Based on these results, it can be assumed that the results shown in Figure 3, and in Table 3 and Table 4, might be improved by optimizing the specimen geometry.

In comparison with 42C steel, the mechanical properties of 42D steel for cooling rates lower than 5 °C/s showed lower values for strength and higher ductile deformation behavior, while for higher cooling rates, the martensite structure for 42D samples was softer and tougher compared with the conventionally prepared 42C and even at a cooling rate of 50 °C/s, the tensile curves of 42D showed a certain plastic deformation prior to fracture. This statement was confirmed by the microhardness measurement HV1 and by metallographic observations and the results are summarized in Table 5 and Figure 6. For a cooling rate of 50 °C, the 42C steel achieved a significantly finer martensite microstructure with higher hardness, namely (755 ± 6) HV1 compared with the 42D, that implies higher embrittlement without any additional heat treatment and, therefore, premature fracturing of tested samples occurred.

In the CCT diagram for H13 steel in Figure 5, it can be seen that this steel had undergone full martensite transformation for all cooling rates presented. The microstructure consisted of martensite matrix and fine carbides for all cooling rates (M + C symbolized in Figure 5). Grain size in Figure 5 refers to the grain size of austenite that existed prior to its transformation. The austenite grain size is stated in the CCT diagrams because the austenite grain size influences the martensite transformation through the density of nuclei at the grain boundaries. It is assumed that an increase in cooling rate increases the dislocation density in the martensite and refines the martensite lath which leads to increase hardness. The MTT results showed increasing strength and decreasing ductility with increasing cooling rate. For a cooling rate of 100 °C/s, brittle behavior characterized by almost no observed elongation and fracture before reaching a maximum load was observed. The microhardness measurement, and light metallography especially, did not find any differences in the microstructures for individual cooling rates to reveal more a brittle behavior of the material with increasing cooling rates, so the fractography was performed to confirm that with the decreasing cooling rate the material became tougher.

Fractography analysis of the steel H13 proved a mixed mode trans-granular fracture. Electron micrographs are shown in Figure 7. The fracture morphology of the tensile-tested surfaces exhibited dimple morphology along the edges of the samples at cooling rates from 100 to 1 °C/s, whereas the quasi-cleavage facets and scattered dimples covered the majority of the fracture surfaces. The proportion of quasi-static fractures, i.e., combination of ductile and brittle fracture mechanisms, increased with the increasing cooling rate and it filled almost the entire area of fracture for the sample cooled at a rate of 100 °C/s. Initiation positions of breakage were located either at the lack of fusion defects (Figure 7a,c) or were caused due to the presence of intergranular disruption of the material (Figure 7a,b). These defects (such as voids, pores) might come from unoptimized processed parameters (e.g., low laser power, high speed, etc.).

## 5. Conclusions

Mini-tensile tests were successfully used to determine the mechanical properties from small round samples (volume of 1.4 × 10^−7^ m^3^) subjected to dilatometry to supply continuous cooling transformation (CCT) diagrams for steel 42CrMo4 conventionally manufactured and directed energy deposited and for H13 tool steel prepared with selective laser melting.The comparisons of MTT with STT were presented for commercially available steel for quenching 42CrMo4. MTT curves revealed almost identical curve shapes as the standard test sample.Using the mini-tensile test, standard CTT diagrams were extended with the information about the strength, ductility and estimated value of the work-hardening coefficient with no need of additional material.For 42CrMo4 steel prepared conventionally and with the DED process, the phase transformation temperatures for individual cooling rates did not differ from each other by more than 10%. The mechanical properties of DED-prepared samples showed lower values for strength and higher ductile deformation behavior.H13 steel underwent full martensite transformation and the fractography demonstrated that with the decreasing cooling rate the material exhibited higher ductile behavior. Neither light microscopy observation nor the microhardness measurement revealed brittle behavior and information about the elongation of the material could not be supplied.

## Figures and Tables

**Figure 1 materials-15-03076-f001:**
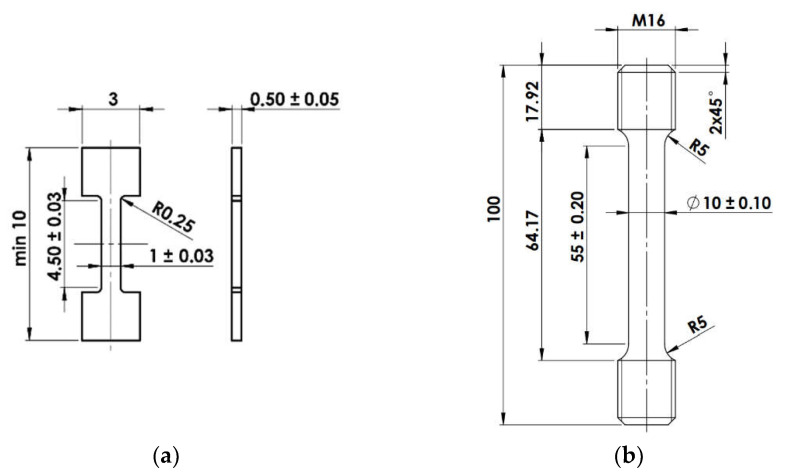
Specimen geometry used for (**a**) mini-tensile test (**b**) standard tensile test.

**Figure 2 materials-15-03076-f002:**
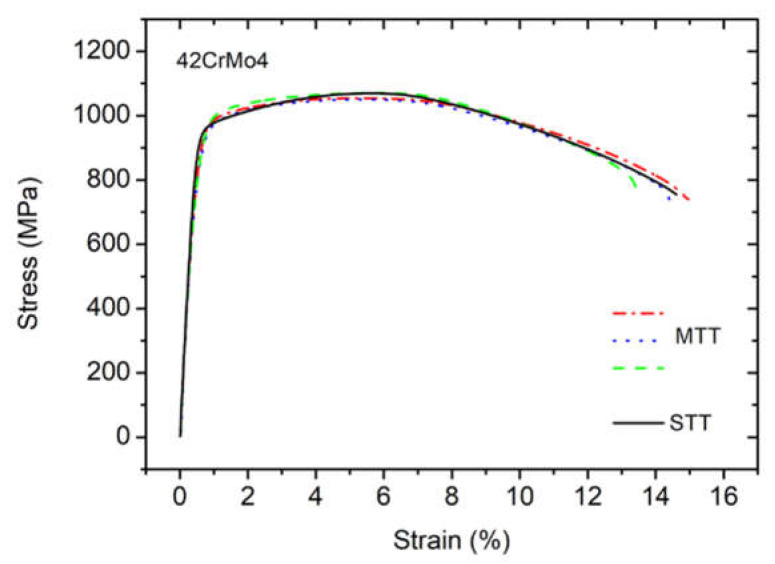
Tensile test results for standard tensile (STT: solid line) and mini-tensile tests (MTT: discontinuous lines).

**Figure 3 materials-15-03076-f003:**
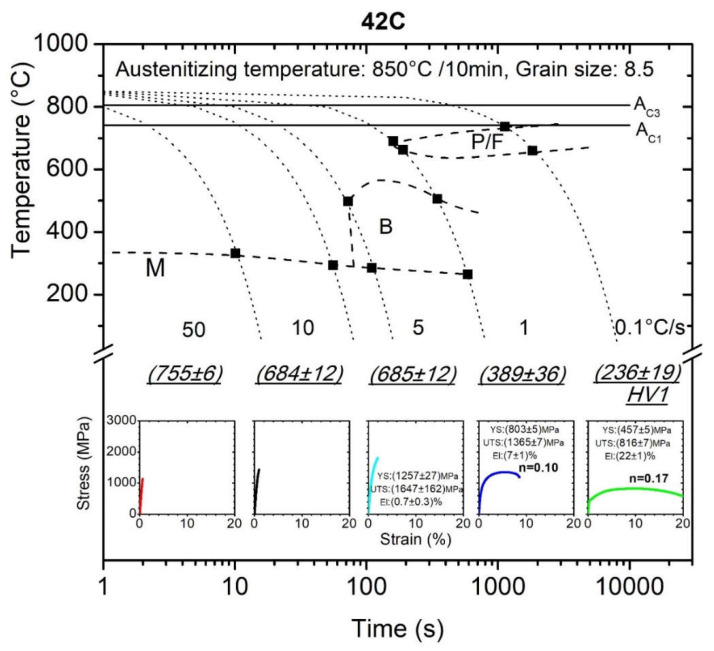
CCT diagram for conventionally manufactured 42C (P/F: perlite + ferrite, B: bainite, M: martensite).

**Figure 4 materials-15-03076-f004:**
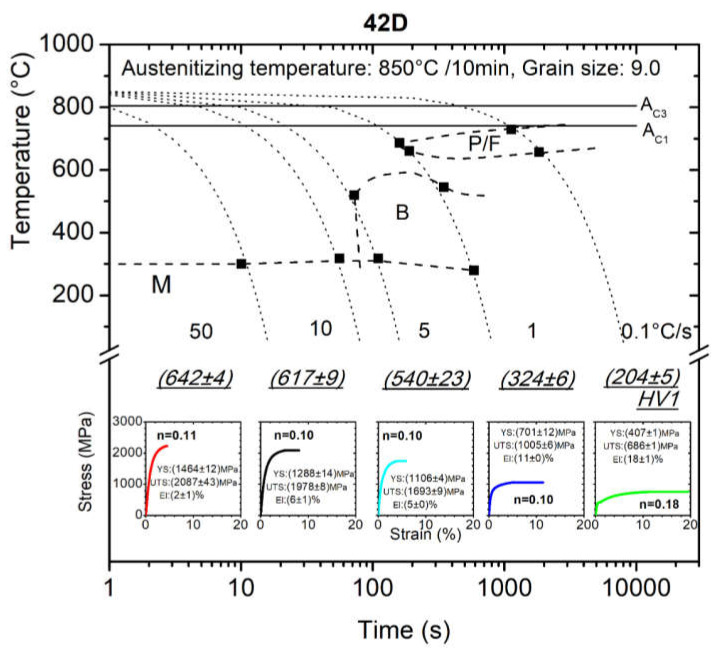
CCT diagram for 42D steel (P/F: perlite + ferrite, B: bainite, M: martensite).

**Figure 5 materials-15-03076-f005:**
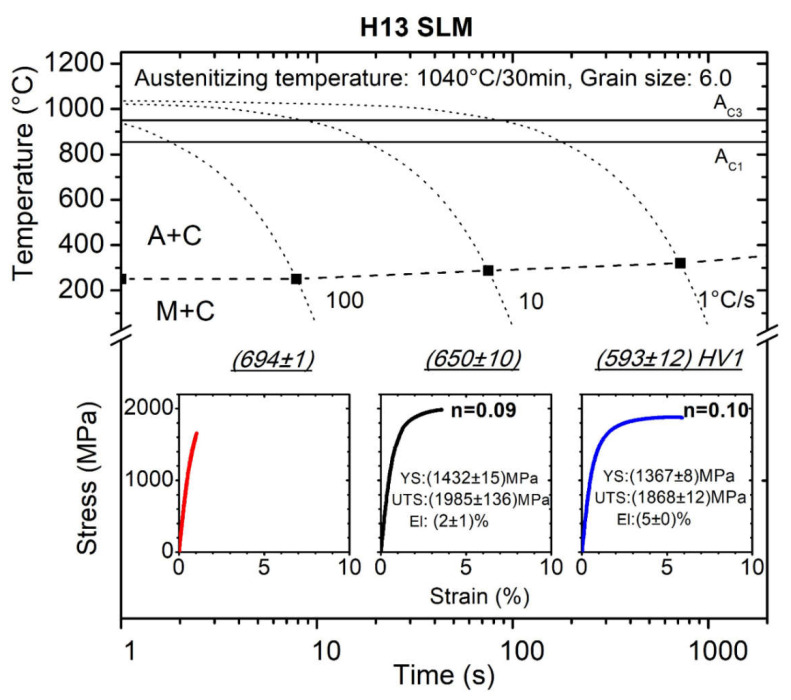
CCT diagram for SLM prepared H13 (A + C: austenite + carbide, M + C: martensite + carbide).

**Figure 6 materials-15-03076-f006:**
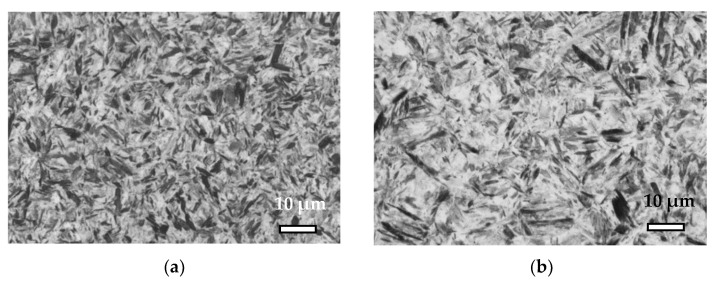
(**a**) Microstructure of 42C steel for cooling rate of 50 °C/s; (**b**) microstructure of 42D steel for cooling rate of 50 °C/s.

**Figure 7 materials-15-03076-f007:**
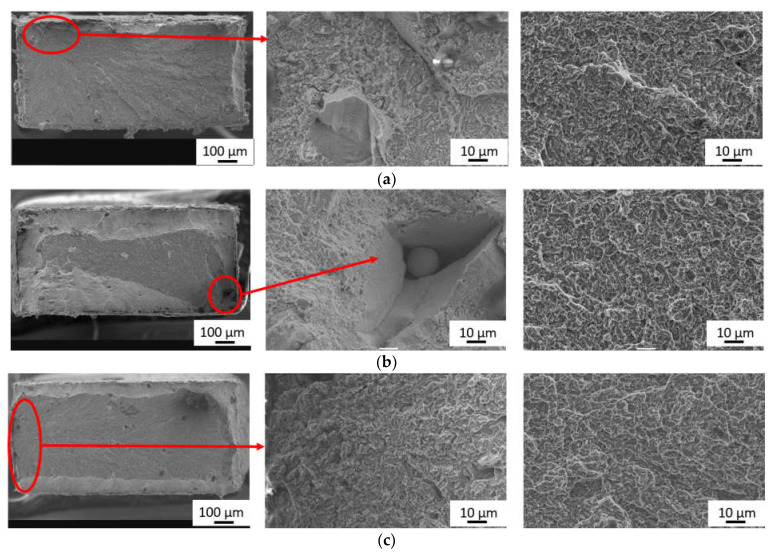
Fracture surfaces of H13 steel after tensile test for cooling rate of: (**a**) 100 °C/s; (**b**) 10 °C/s; (**c**) 1 °C/s.

**Table 1 materials-15-03076-t001:** Chemical composition of presented steels.

Material	wt%													
	C	Mn	P	S	Si	Cr	Mo	Ni	Cu	V	Al	Nb	Ti	Fe
**42C**	0.41	0.82	0.02	0.02	0.25	1.05	0.23	0.51	0.05	0.02	<0.01	<0.01	<0.01	Bal.
**42D**	0.40	0.87	0.035	0.04	0.25	0.95	0.20	-	-	-	-	-	-	Bal.
**H13**	0.42	0.44	0.01	0.01	0.85	5.22	1.50	0.01	0.01	1.04	0.01	<0.01	<0.01	Bal.

**Table 2 materials-15-03076-t002:** Summary of phase transformation temperatures for individual cooling rates measured for 42C and 42D steel. (Ms: martensite start temperature, Mf: martensite finish temperature, B: bainite, P: perlite).

Material	Temperature (°C)		Cooling Rate (°C/s)				
	Ac1	Ac3	50	10	5	1	0.1
**42C**	740	805	Ms = 330 °C	Ms = 295 °C	Bs = 500 °C	Ps = 690 °C	Ps = 736 °C
					Ms = 290 °C	Pf = 660 °C	Pf = 660 °C
						Bs = 510 °C	
						Ms = 270 °C	
**42D**	740	805	Ms = 300 °C	Ms = 320 °C	Bs = 520 °C	Ps = 690 °C	Ps = 730 °C
					Ms = 320 °C	Pf = 660 °C	Pf = 655 °C
						Bs = 545 °C	
						Ms = 280 °C	

**Table 3 materials-15-03076-t003:** Results of mechanical properties determined from mini-tensile tests for 42C according to the experimental cooling rates (YS: yield strength, UTS: ultimate tensile strength, El: total elongation at fracture, *n*: work-hardening exponent).

Material	Cooling Rate (°C/s)	YS (MPa)	UTS (MPa)	El (%)	*n* (-)
**42C**	0.1	457 ± 5	816 ± 7	22 ± 1	0.17
	1	803 ± 5	1365 ± 7	7 ± 1	0.10
	5	1257 ± 27	1647 ± 162	0.7 ± 0.3	-
	10	brittle behaviour			-
	50				-

**Table 4 materials-15-03076-t004:** Results of mechanical properties determined from mini-tensile tests for 42D steel according to the experimental cooling rates (YS: yield strength, UTS: ultimate tensile strength, El: total elongation at fracture, *n*: work-hardening exponent).

Material	Cooling Rate (°C/s)	YS (MPa)	UTS (MPa)	El (%)	*n* (-)
**42D**	0.1	407 ± 1	686 ± 1	18 ± 1	0.18
	1	701 ± 12	1005 ± 6	11 ± 0	0.10
	5	1106 ± 4	1693 ± 9	5 ± 0	0.10
	10	1288 ± 14	1978 ± 8	6 ± 1	0.10
	50	1464 ± 12	2087 ± 43	2 ± 1	0.11

**Table 5 materials-15-03076-t005:** Microhardness HV1 measurement for 42C and 42D steels.

Material	Cooling Rate (°C/s)				
	0.1	1	5	10	50
	HV1				
**42C**	236 ± 19	389 ± 36	685 ± 12	684 ± 12	755 ± 6
**42D**	204 ± 5	324 ± 6	540 ± 23	617 ± 9	642 ± 4

## Data Availability

The raw data required to reproduce these findings cannot be shared at this time as the data also forms part of an ongoing study.

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
