# Peer review of "Extended Continuous Cooling Transformation (CCT) Diagrams Determination for Additive Manufacturing Deposited Steels"

_materials, 2022, doi:10.3390/ma15093076_

Round 1
Reviewer 1 Report
The authors present a comparative study that investigates the cooling transformation characteristics of three kinds of steels processed by different treatment and their tensile properties and hardness, and as well try to argue that the mini tensile testing is an available alternative approach if the raw material lacks. Before being accepted for publication, the following modification would be strongly recommended and carefully considered.
1 Checking whether the units and mathematical symbols in the original manuscript are correct and revising them correctly.
2 The results and discussion section should be divided into subsections to improve the structure of the article.
3 The conclusion would be listed by point.
4 The paragraph on lines 242-244 does not seem to be the content of this article.
Author Response
The response to your comments was uploaded as the PDF file.

Reviewer 2 Report
Comments to the manuscript “Extended continuous cooling transformation (CCT) diagrams determination for additive manufacturing deposited steels” by Kristýna Halmešová et al.
The authors constructed the CCT diagrams of 42CrMo4 steel that was prepared by the SLM on the basis of dilatometry measurements. Indeed, I could not find anything new beyond present knowledge of the 42CrMo4 steel. This manuscript is more like to provide a CCT diagram data of the SLM steel to readers. The MTT used in the manuscript is a better method to evaluate the mechanical properties of materials after cooled at exactly controlled rates than only considering the micro hardness. Beside this no more new theoretical understanding about 42CrMo4 steel can be found. Thus I suggest rejection of the manuscript.
Author Response

(The authors gave the same response as above.)

Reviewer 3 Report
- The paper seems to be interesting, however, in the current state the novelty of the work is not evident. For example, it is naturally to use for two procedures the same specimens if their size meets requirements in both cases. Given this the novelty of this work is problematical. In contrast to this, the comparison of tensile strength data obtained for the selected materials with the use of standard sized and MTT specimens could be main objective of the paper.
- Chemical composition of all the studied steels should be obligatorily provided in the section Materials and Methods.
- The authors should provide information how many specimens were used for each cooling rate!
- The authors should provide information how many measurements were done for each sample and the averaged values should be shown!
- The quality of the micrographs is very poor. Moreover, the microstructures have not been studied and any description or discussion is lacking in the manuscript!
- M, B, P/F, Grain size is not defined in Fig. 2, 3 and 5, which should be clarified in the text or in the captions for the corresponding figures because these can be unknown for some categories of readers!
- Bad quality of micrographs in the mentioned figures, as well as in Fig. 4!
- Standard deviation should be shown everywhere!
- It is not clear why fractography was carried out only for one steel and how this test relates to the main aim of the paper. What is the significance of this test? In the conclusions, nothing was said about the results of this test!
All the detailed comments are pointed out in the original manuscript attached.

Author Response

(The authors gave the same response as above.)

Round 2
Reviewer 2 Report
The manuscript has been revised adequately. I agree with the publication of the manuscript on the journal.
Author Response
The response was uploaded as PDF.

Reviewer 3 Report
The authors have partially revised and improved their manuscript. However, without substantial metallographic support or more deep investigations, for example with respect to the fractography of the mini-tensile test specimens compared with conventional ones the technical and scientific content of the paper is not sufficient for such highly impacted journal as Materials.
Author Response
The response was uploaded as PDF.
